# Learning then Leveraging Structures Help with Complex, Compositional, Causal Sequential Tasks in Inverse Reinforcement Learning

## Abstract

The field of Inverse Reinforcement Learning (IRL) has experienced substantial advancements in recent years, with commendable newer approaches yielding crucial applications in diverse areas such as robotics, cognition, and healthcare. This paper underscores the limitations of foundational IRL methods when learning an agent's reward function from expert trajectories that have underlying causal reward structure. We posit that imbuing IRL models with causal structural motifs capturing underlying relationships between actions and outcomes or, the reward logic can enable and enhance their performance. Based on this hypothesis, we propose SMIRL – an IRL approach that initially learns the task's structure as a finite-state-automaton (FSA) and subsequently leverages this structural motif to solve the IRL problem. We demonstrate SMIRL's capabilities in both discrete (grid world) and high-dimensional continuous domain environments across four logic based tasks. The SMIRL approach proves adept at learning tasks characterized by causal reward functions, a known limitation of foundational IRL approaches. Our model also outperforms the baselines in sample efficiency on tasks. We further show promising test results in a modified continuous domain on tasks with compositional reward functions.

## 1 Introduction

Inverse Reinforcement Learning (IRL) coupled with *Behaviour Cloning* is a dominant approach under the broad category of learning methods dubbed as *Imitation Learning*, where agents learn control policies by observing and imitating the behaviour of a teacher or expert demonstrator (Ng et al., 2000; Pomerleau, 1991; Schaal, 1999). Unlike Behaviour Cloning, where the agent directly learns a policy by mapping observed states to actions, IRL entails learning the underlying reward function of the demonstrator based on observations and a model of the environment, subsequently using the learned reward function to derive policy. IRL has evolved considerably since coined and introduced by Russell (1998) and has been applied in various different contexts and domains, including robotics, cognition, health (Argall et al., 2009; Baker et al., 2009; Asoh et al., 2013).

The early works in IRL entailed representing the reward function as a weighted linear combination of handcrafted features Abbeel & Ng (2004). Essentially, a strategy of matching *feature expectations* between an observed policy and an agent's behaviour Arora & Doshi (2021). However, such approaches have inherent weaknesses in delineating trajectories from sub-optimal policies. To exemplify, consider the canonical degenerate case of all zeroes. More recent approaches Ziebart et al. (2008); Wulfmeier et al. (2015); Ni et al. (2020); Chan & van der Schaar (2021) circumvent this weakness using max-entropy Jaynes (1957) based

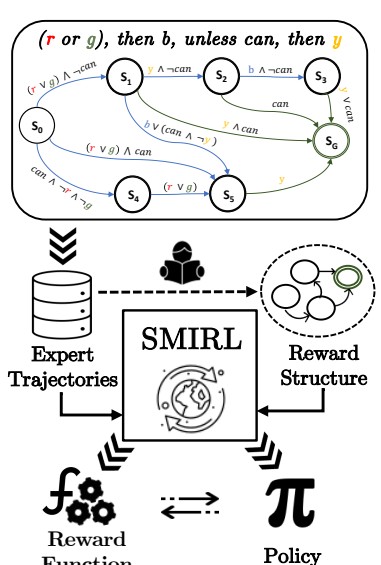

Figure 1: A conceptual schematic diagram of the SMIRL approach.

principled approaches that consider a distribution over all possible behaviour trajectories and favour the trajectories with less ambiguity.

While these approaches show demonstrable success over tasks formulated as MDP problems Adams et al. (2022); Arora & Doshi (2021), in this work we explore their efficacy in sequential tasks with underlying causal structure, where the rewards received by an agent are **not necessarily Markovian** with respect to the state space, and current actions cause future observations. De Haan et al. (2019) demonstrated the "causal misidentification" phenomenon emanating from ignoring causality, or when learning procedures are unaware of the causal structure of interactions between an expert and the environment, broadly within *Imitation Learning* domain. Here we specifically focus on examining approaches based on IRL.

As such, we conduct experiments first on a simple 2D discrete environment (Section 4.2), followed by a high-dimensional continuous domain environment (Section 4.3) using tasks with varying complexity between observations and underlying state reward. We mimic causal structure among states' propagation with logical conditions Vardi (1996); Giacomo & Vardi (1999). To elucidate, a simple MDP problem of moving to a specific location on a map (**Task 0**) [Figure 2a] were modified with more challenging long-horizon tasks (**Tasks 1-4**) [Figure 2c], like patrolling a set of locations in conditional (e.g., sequential) orders to receive rewards. We found that while both baselines succeeded in the simple task, they completely failed on the harder tasks.

We hypothesize that imbuing IRL models with a *structural motif* of the underlying complex task can enable the models to learn the reward functions even in such long-horizon complex causal scenarios. Based on our hypothesis, we propose a novel IRL model **SMIRL** (Figure 1) that learns a finite-state-automaton (FSA) based causal reward structure as motif, then uses the structural motif in solving the IRL problem. We empirically show that our proposed model is able to successfully learn four complex sequential tasks in the *Office Grid* environment where the other baselines using foundational IRL approaches (Apprenticeship Abbeel & Ng (2004), MaxEnt (Ziebart et al., 2008; Wulfmeier et al., 2015)) categorically fail.

## 2 PRELIMINARIES AND BACKGROUND

**Markov Decision Process (MDP)**   A MDP is defined by a tuple $(\mathbb{S}, \mathbb{A}, T, R, \gamma, p_0)$ where $\mathbb{S}$ is a set of states (state space), $\mathbb{A}$ is a set of actions (action space), $T : \mathbb{S} \times \mathbb{A} \to \Pi(\mathbb{S})$ is the transition function, $R : \mathbb{S} \to \mathbb{R}$ is the reward function, $\gamma \in [0, 1]$ is the discount factor, and $p_0 : \mathbb{S} \to [0, 1]$ is the distribution over initial states. A policy over a MDP is a function $\pi : \mathbb{S} \to \Pi(\mathbb{A})$, and is optimal if it maximizes the expected discounted sum of rewards.

**Partially Observable Markov Decision Process (POMDP)**   A POMDP is a generalisation of a MDP defined by the tuple $(\mathbb{S}, \mathbb{A}, T, \mathbb{O}, \omega, R, \gamma, p_0)$ where $O$ is a set of observations and $\omega : \mathbb{S} \to \Pi(\mathbb{O})$ is the observation function. An agent in a POMDP thus only receives an observation (i.e., partial information about the state) rather than the actual state of the environment. Therefore, policies on POMDP act based on the history of observations received and actions taken at timestep $t$. Since using the complete history is impractical, many algorithms instead use a **belief** $b : \mathbb{O} \to \Pi(\mathbb{S})$, which is a probability distribution over possible states updated at each timestep. The belief update after taking the action $a \in A$ and receiving observation $o \in \mathbb{O}$ is done through the following equation:

$$b_o^a(s') = P(s' \mid b, a, o) = \frac{\omega(s', o) \sum_s T(s, a, s') b(s)}{P(o \mid b, a)} \ \ \forall s' \in S, \tag{1}$$

where $P(o \mid b, a) = \sum_{s'} \omega(s', o) \sum_s T(s, a, s') b(s)$.

**Inverse Reinforcement Learning (IRL)**   Given some demonstrations $\mathcal{D} = \{\tau_1, \tau_2, \ldots, \tau_N\}$ generated by an expert policy $\pi_E$, the goal of inverse reinforcement learning is to learn a reward function that can explain the behaviour of the expert policy (Ng et al., 2000; Abbeel & Ng, 2004; Arora & Doshi, 2021; Konar et al., 2021). Unlike *Imitation Learning* Hussein et al. (2017), learning reward functions can provide better generalizations of the agent when it comes to a new environment compared to only mimicking expert behaviours.

**IRL Approaches**   We use two popular and foundational approaches from the dense IRL literature Arora & Doshi (2021) as our baselines. '*Apprenticeship Learning*' Abbeel & Ng (2004) is a seminal work and still relevant today Escontrela et al. (2022). This approach entails calculating

feature expectations for the states of the expert and the apprentice (learner) trajectories separately, then optimizing the reward function parameters using *max-margin* or *quadratic programming* (QP). The learner's feature expectations change along the learning process. An iterative procedure is then usually used to update the learner's policy and the reward function for each state representation, until the algorithm converges.

Unfortunately, this approach of matching feature counts between expert and learner is ambiguous. Sub-optimal policies can lead to the same feature counts (including the degenerate cases, e.g., all zeroes). The **Maximum Entropy** IRL Ziebart et al. (2008) approach deals with this ambiguity in a principled way using the eponymous maximum entropy principle Jaynes (1957). Broadly, this approach considers a distribution over all possible behaviour trajectories, and disfavours ambiguous policies with high entropy, choosing the least ambiguous policy that does not exhibit additional information beyond matching feature expectations. The resulting distribution over trajectories ($\tau_i$) for deterministic MDPs is parameterized by reward weights $\theta$, such that, trajectories with equivalent rewards have equal probabilities, and trajectories with higher rewards are preferred exponentially more (Eq. 2):

$$P(\tau_i \mid \theta) = \frac{1}{Z(\theta)} e^{(\theta^T \mathbf{f}_{\tau_i})} \tag{2}$$

where $Z(\theta)$ is the partition function, $\tau$ is the trajectory, and $f_{\tau_i}$ is the feature representation.

**IRL with Partial Observability**  Choi & Kim Choi & Kim (2011), extending their earlier work Choi et al., tackled the problem of IRL in partially observable domains (POMDP). The authors introduced algorithms in two settings – one where the expert policy is explicitly given, and one where only the expert trajectories are available. The latter is relevant to our work here. However, unlike our algorithm, introduced in Section 3, the authors assumed full access to the environment transition probabilities in order to compute the trajectory of beliefs using Eq. 1. Once such a trajectory is obtained, usual IRL algorithms can be applied to the belief-state MDP.

**Reward Machine (RM)**  Introduced by Icarte et al. (2018), RM is a standardized ontology for finite state automatons (FSA) that can be used to define the reward function inside a MDP (Icarte et al., 2020). Given a set of **propositional symbols** $\mathcal{P}$ (which are linked to the environment states through a **labelling function** $L : S \to 2^{\mathcal{P}}$), a reward machine (RM) is defined by a tuple $(\mathbb{U}, u_0, \delta_u, \delta_r)$, where $\mathbb{U}$ is a finite set of states, $u_0 \in \mathbb{U}$ is the initial state, $\delta_u : \mathbb{U} \times 2^{\mathcal{P}} \to \mathbb{U}$ is the **state-transition function** and $\delta_r : \mathbb{U} \to [\mathbb{S} \to \mathbb{R}]$ is the **state-reward function**. In other words, the RM's state is updated at each time-step through $\delta_u$ and the reward that the agent gets from the environment is defined by the reward function output by $\delta_r$.

It is important to note that RMs can express temporally extended tasks, which are therefore non-Markovian. However, we can define an MDP over the cross-product $\mathbb{S} \times \mathbb{U}$, which keeps its Markovian property (Meuleau et al., 2013a). Expressing the reward function in such a way allows an RL agent to decompose a task into structured sub-problems and learn them effectively. The Q-learning for RM (QRM) algorithm simultaneously learns a separate policy, $\tilde{q_u}$, for each state ($u \in \mathbb{U}$) in the RM using Q-learning Icarte et al. (2018). In subsequent work, the **LRM** algorithm learns RM from a set of trajectories using discrete optimization Toro Icarte et al. (2019). The learned reward can then be used to ease the learning in the environment, especially in POMDPs. The authors notably showed that if a POMDP has a finite belief space, then there exists a RM for an adequate labelling function $L$ such that the POMDP extended with the RM is Markovian with respect to the observations (i.e., the entire history can be explained by only the last observation and the current RM state). Please see Appendix Section C for further discussions on related work and differentiation of our work here.

## 3  Structural Motif-Conditioned IRL (SMIRL)

Given a POMDP $\mathcal{M} = (\mathbb{S}, \mathbb{A}, T, \mathbb{O}, \omega, R, \gamma, p_0)$, we consider the IRL problem of learning a reward function $R_\theta$ given a set of expert trajectories $\mathcal{D} = \{\tau_i\}$ where $\tau_i = (o_0, a_0, o_1, a_1, ..., a_T, o_T)$, such that an optimal policy for $\mathcal{M}_\theta = (\mathbb{S}, \mathbb{A}, T, \mathbb{O}, \omega, R_\theta, \gamma, p_0)$ is also optimal for $\mathcal{M}$. We assume that we have access to $\mathcal{M} \setminus \{R\}$ when learning $R_\theta$. Moreover, since we want to use reward machines, we assume that we have access to a relevant set of propositions $\mathcal{P}$ and the corresponding labelling function $L : \mathbb{O} \to 2^{\mathcal{P}}$.

The main idea behind our algorithm (Figure 1) is to first learn a FSA structure $(\mathbb{U}, u_0, \delta_u)$ on the POMDP using the trajectories in $\mathcal{D}$, and then use **MaxEnt IRL** (Ziebart et al., 2008) on the resulting

---

**Algorithm 1:** SMIRL

---

**Input:** Set of expert trajectories $\mathcal{D}$

Learn a FSA structure $(\mathbb{U}, u_0, \delta_u)$;

Update the trajectories in $\mathcal{D}$ with the corresponding RM states;

**for** $(u, o) \in \mathbb{U} \times \mathbb{O}$ **do**
$\quad \mid \quad \mu_{\mathcal{D}'}(u, o) \leftarrow \sum_{\tau' \in \mathcal{D}'} \sum_{o_t, u_t, a_t \in \tau'} \mathbf{1}_{u=u_t} \mathbf{1}_{o=o_t}$
**end**

Initialize $\theta$;

**for** $m = 1$ *to* $M$ **do**
$\quad$ Compute $\pi_\theta$ optimal policy over $\mathcal{M}_{RM,\theta}$;
$\quad$ **for** $(u, o) \in \mathbb{U} \times \mathbb{O}$ **do**
$\quad \quad \mid \quad \mu_{\pi_\theta}(u, o) \leftarrow \mathbb{E}_{\tau \sim \pi_\theta} \left( \sum_{o_t, u_t \in \tau} \mathbf{1}_{u=u_t} \mathbf{1}_{o=o_t} \right)$
$\quad$ **end**
$\quad$ Compute $\frac{\partial L}{\partial \theta} = \sum_{u,o \in \mathbb{U} \times \mathbb{O}} (\mu_{\mathcal{D}'}(u, o) - \mu_{\pi_\theta}(u, o)) \frac{\partial \delta_{r,\theta}(u,o)}{\partial \theta}$;
$\quad$ Update $\theta$;
**end**

---

extended POMDP to learn the RM's state-reward function $\delta_{r,\theta}$, where $\theta$ represents the learnable parameters of the function.

**Learning the Reward Machine** In order to learn the RM's structure, we build on the method described in Toro Icarte et al. (2019), where *Tabu search* Glover & Laguna (1998) is used to compute an RM that minimize the following objective:

$$\operatorname*{argmin}_{\langle U, u_0, \delta_u \rangle} \sum_{\tau \in \mathcal{D}} \sum_{o_t, a_t \in \tau} \log \left( \left| N_{x_{\tau,t}, L(o_t)} \right| \right), \tag{3}$$

where $x_{\tau,t}$ refers to the state of the RM at timestep $t$ when rolling out trajectory $\tau$, and $N_{u,l} \in 2^{2^{\mathcal{P}}}$ is the set of all the next abstract observations (i.e., outputs of the labelling function $L$) seen from the RM state $u$ and the abstract observations $l$. In other words, $l' \in N_{u,l}$ iff $u = x_{\tau,t}$, $l = L(o_t)$ and $l' = L(o_{t+1})$ for some trace $\tau$ at timestep $t$. The search is further constrained by imposing a limit $u_{max}$ to the number of states of the RM.

Once the FSA structure is learned, we can update the trajectories in $\mathcal{D}$ with the corresponding RM states, i.e., $\mathcal{D}' = \{\tau_i'\}$ where $\tau_i' = (o_0, u_0, a_0, o_1, u_1, a_1, ..., o_T, u_T)$. We assume for the rest of this section that the learned RM is 'perfect' (meaning that there is a MDP defined on the state space $\mathbb{U} \times \mathbb{O}$, denoted $\mathcal{M}_{RM}$, such that any optimal policy for $\mathcal{M}_{RM}$ is optimal for $\mathcal{M}$. We thus can learn a reward function on $\mathcal{M}_{RM}$ instead of $\mathcal{M}$.

**Learning the Reward Function**. As $\mathcal{D}'$ can be seen as a set of trajectories in $\mathcal{M}_{RM}$, we can now use IRL algorithms that apply to MDPs such as MaxEnt IRL to learn the reward function, that we now denote $\delta_{r,\theta} : U \times O \rightarrow \mathbb{R}$. MaxEnt IRL assumes a maximum entropy constraint on the distribution of possible trajectories. In our case, this translates to:

$$P_\theta(\tau') \propto \exp \left( \sum_{o_t, u_t, a_t \in \tau'} \delta_{r,\theta}(u_t, o_t) \right) \tag{4}$$

An optimal reward function can then be learned by maximizing the log-likelihood $L$ of the expert trajectories in $\mathcal{D}'$:

$$\theta^* = \operatorname*{argmax}_\theta L(\theta) = \operatorname*{argmax}_\theta \sum_{\tau' \in \mathcal{D}'} \log P_\theta(\tau'). \tag{5}$$

As shown in Wulfmeier et al. (2016), the above $\theta^*$ can be computed by gradient ascent using:

$$\frac{\partial L}{\partial \theta} = \sum_{u,o \in U \times O} (\mu_{\mathcal{D}'}(u, o) - \mu_{\pi_\theta}(u, o)) \frac{\partial \delta_{r,\theta}(u, o)}{\partial \theta}, \tag{6}$$

where $\mu_{\mathcal{D}'}$ is the *state visitation frequency* (SVF) of the expert demonstrations and $\mu_{\pi_\theta}$ is the SVF of the policy optimal with respect to $\mathcal{M}_{RM,\theta}$ (i.e., $\mathcal{M}_{RM}$ with the learned reward function $\delta_{r,\theta}$).

It is common practice in IRL to apply the reward function to handcrafted features $\phi : S \to \mathbb{R}^N$ instead of directly using the raw state (Arora & Doshi, 2021; Konar et al., 2021). In our experiments (Section 4), we use the labelling function $L : O \to 2^{\mathcal{P}}$ as the feature function. The reward at state $u, o \in U \times O$ is therefore $\delta_r(u, L(o))$. We further assume that the reward function is linear with regards to $L(o)$ (with $L(o) \in \{0, 1\}^{|\mathcal{P}|}$), i.e.,

$$\forall u \in U, \ \exists \, \mathbf{w}_u \in \mathbb{R}^{|\mathcal{P}|} \ \text{ s.t. } \ \delta_r(u, L(o)) = \mathbf{w}_u^\top L(o) \tag{7}$$

## 4 EXPERIMENTS

### 4.1 BASELINES

**Apprenticeship-IRL** For the discrete *Office Gridworld* experiments, we use the classical *Apprenticeship* IRL model (Abbeel & Ng, 2004) for both inferring reward function and learning an optimal agent from given expert trajectories, $\mathcal{D}_e$. The initial step entails learning the environment's high-level feature expectations using Monte Carlo estimate on the $\mathcal{D}_e$ samples for the expert and simulated trajectories by an initialized policy, $\pi_l$ (e.g., a Q-table) for the learner.

$$\mu(\pi) = \mathbb{E}_\tau \left[ \sum_{t=0}^{\infty} \gamma^t s_t^h \bigg| \pi \right] \approx \sum_{i=1}^{n} \sum_{t=0}^{\infty} \gamma^t s_t^h \in \mathbb{R}^9 \tag{8}$$

where $s_t^h$ represents the high-level features at time step $t$. The outer layer of summation is over different trajectories, and the inner layer of summation is over time steps within one trajectory.

As per the IRL definition, the goal is to learn a weight $\mathbf{w}$ such that the reward can be modelled as a weighted linear function of the high-level state features $R(s_t) = \mathbf{w}^\intercal s_t^h$. Quadratic programming (QP) optimization Floudas & Visweswaran (1995) is applied to the feature expectations from Eq. 8 to find a minimal scale weight parameter such that the constraint is satisfied (Eq. 13 from Abbeel & Ng (2004):

$$\begin{aligned} QP(\mathbf{w}) = min \quad & ||\mathbf{w}|| \\ s.t. \quad & [\mu(\pi_e) - \mu(\pi_l)]^\intercal \mathbf{w} \geqslant \epsilon \end{aligned} \tag{9}$$

where $\mu(\pi_e), \mu(\pi_l)$ are feature expectations of the expert and learner trajectories respectively, and $\epsilon$ is a pre-defined hyperparameter. After getting an initialized reward function from the QP optimization process, the Q-function is learned through the general temporal difference learning Sutton & Barto (2018) process.

**MaxEnt-IRL** The second baseline uses another foundational IRL method introduced in Ziebart et al. (2008), that uses maximum entropy based approach for deducing the reward function from expert trajectories. The difference with our approach is that we apply the algorithm to the observations, $O$ without conditioning on the structural motif, $U \times O$.

**f-IRL** To test where our approach can learn good policies in high-dimensional continuous control tasks, we use the f-IRL approach Ni et al. (2020) as a baseline that learns a reward function and optimal policy simultaneously using *state marginal matching*.

### 4.2 OFFICE GRIDWORLD

We use the discrete state '*Office Gridworld*' environment used in Icarte et al. (2018) for our initial experiments. We designed five tasks (referred to as '*Task n*' where $n \in [0 - 4]$ ) on this domain. Fig. 2 shows the schematic diagram and examples of task specific finite state automatons (i.e. RMs) in this domain.

**Tasks** The first task (*Task 0*) is a simple MDP task designed to verify the efficacy of our proposed algorithm and the foundational IRL baselines. With a single destination location D on the map, the goal for the agent is to reach D without going over any obstacle. The other four tasks are sequential non-Markovian tasks used in Icarte et al. (2018). These tasks can be dichotomized into *delivery* and *patrol* tasks. In **Task 1** (resp. **Task 2**), the agent is expected to fetch and deliver coffee (resp. the mail) to the office. In **Task 3**, the agent is expected to deliver both. **Task 4** requires the agent to patrol or visit the grid in a specified sequential route: $A \to B \to C \to D$ in order to receive a reward. Figures 2b, 2c show the perfect RM structures that describes tasks 3 & 4.

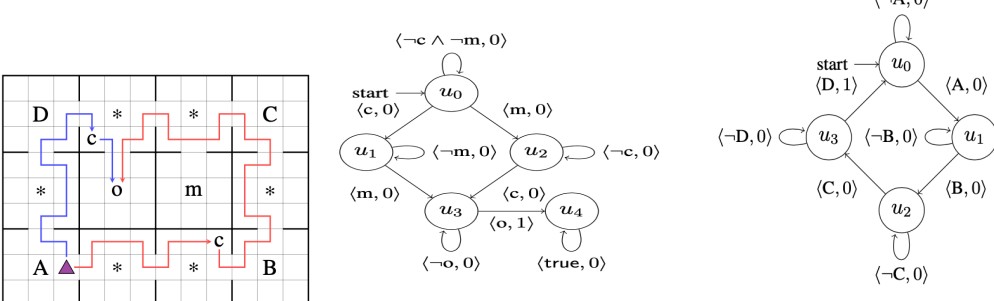

(a) The Office GridWorld

(b) Tasks 1-3: Delivering Coffee & Mail

(c) Task 4: Patrol A,B,C, and D

Figure 2: **The Office GridWorld environment** (images from Icarte et al. (2018)) . In **(2a)**: the small pink triangle shows the agent's position. The small letters o,c,m shows the location of the office, coffee, and mail respectively. The capital letters A,B,C,D represents the four locations that can be visited by the agent. The asterik * represents an obstacle. The blue and red lines represent the optimal and sub-optimal paths respectively for completing Task 1. Fig. **(2b)**, **(2c)** show ground-truth FSA structure for sub-captioned tasks in the Office GridWorld environment.

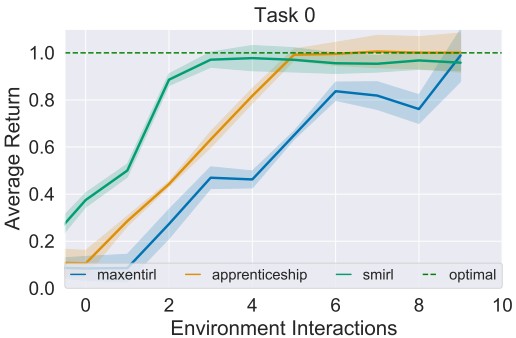

Figure 3: **Single-Step MDP (Task 0)**: reach location D avoiding obstacles. All algorithms succeed in this pedagogical example. A final reward of 1 shows successful task objective completion. Ours (in green) is sample efficient and always takes the optimal (blue) path in Fig. 2a due to the underlying causal structural motif.

**Generating Expert Trajectories** We generate test-time demonstration trajectories based on the implementation from the original QRM paper[1]. We make sure that the saved demonstrations are optimal ones by counting the number of steps before the trajectories end.

The expert demonstrations are represented by series of observation-action transitions, i.e., $(o_1, a_1, o_2, a_2, ..., o_T)$. The actions are represented by single integers from 0 to 3. The observations contain both the raw observation from the environment (the agent's position on the grid) as well as the output of the labelling function. In the *Office Gridworld*, the labelling function $\mathcal{L}$ outputs the label of the tile the agent is on (i.e., 'A' if the agent is on position A). We refer to this *labels* as '*high-level features*' in subsequent parts.

**Performance in Single Step Tasks** Figure 3 shows the results of the simple MDP task (*Task 0*) – where the agent simply needs to reach the location D from its starting position while avoiding the obstacles. The average reward that the learned policy gets from the ground truth reward function is shown at each IRL step (i.e., each time a new reward function is learned). As we can see, all three algorithms are able to learn an optimal policy on this task, although our algorithm learns in fewer steps.

This faster learning can be explained by the terminal (goal) state in the reward machine. We prevent the environment from resetting to obfuscate the real reward function from the agent. Thereby, during training, the environment does not reset upon task completion (reaching D) or 'game over' (going over an obstacle). Thus, our approach learns the optimal or maximum reward state by inferring the goal state (from the learned FSA), while the other algorithms don't. Also, having the causal structural motif always ensure ours to take the optimal path to success unlike the baselines.

---

[1] https://bitbucket.org/RToroIcarte/qrm/src/master/

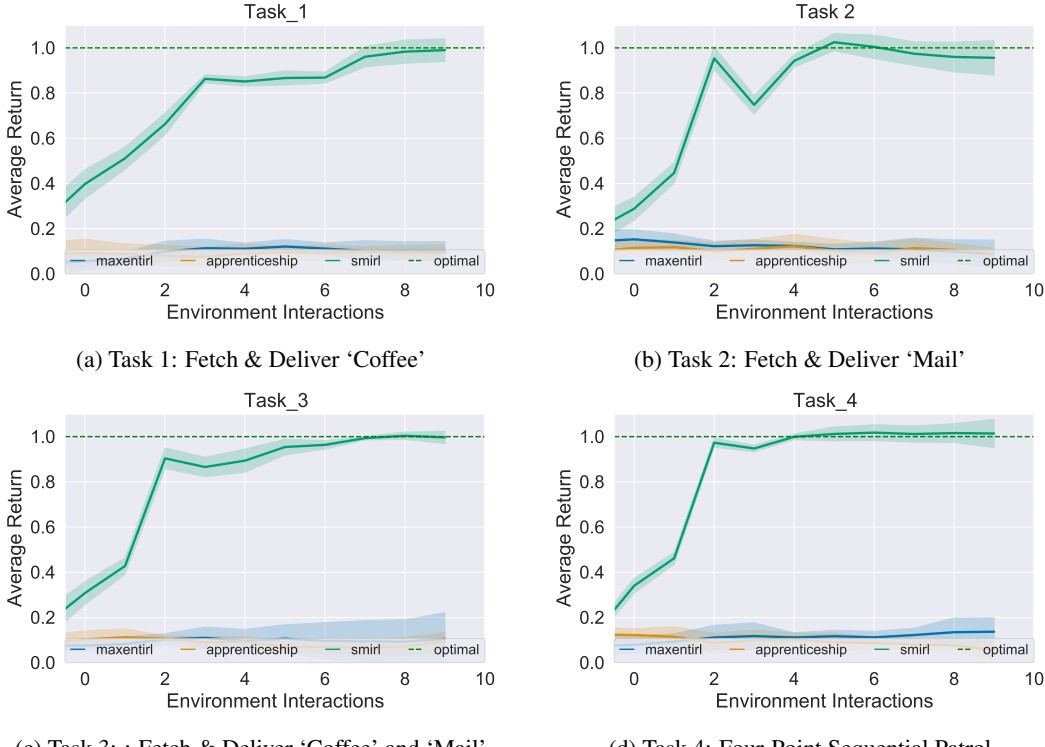

(a) Task 1: Fetch & Deliver 'Coffee'

(b) Task 2: Fetch & Deliver 'Mail'

(c) Task 3: : Fetch & Deliver 'Coffee' and 'Mail'

(d) Task 4: Four-Point Sequential Patrol

Figure 4: **Multi-step Partially-Observable Domains** training curves for SMIRL and baselines on the POMDP tasks in **Office GridWorld** environment. All Environment interactions (x-axis) are $1 \times 10^5$. Reward of 1 is obtained when target location 'o' is reached after satisfying conditional goals without environmental rewards. SMIRL learns the underlying structure with state-transition and reward functions $\delta_u, \delta_r$.

**Performance in Multi-step Partially-Observable Domains (1-4)** However, in the more complicated sequential tasks shown in Fig. 4, only SMIRL succeeds to learn the tasks while the other IRL algorithms fail. This shows that our algorithm is able to learn in a POMDP setting, by learning an adequate causal structure (i.e. a RM) from the expert trajectories. An example learned RM can be seen in Fig. 8b. In these tasks, an reward of 1 is given only when a conditional logic has been met prior to target goal. To elucidate, the delivery tasks (Tasks 1-3) require the agent reaching coffee or mail or both locations prior to reaching the target location: office 'o'. Thus, from the $\mathcal{D}_e$ trajectories, policies observe the reward after reaching 'o', the dependency on the prior conditions make these tasks complex of POMDP nature. Here, ours learn the underlying structure, and corresponding state-transition reward function $\delta_r : \mathbb{U} \to [\mathbb{S} \to \mathbb{R}]$ that allows succeeding at these tasks.

## 4.3 REACHER DOMAIN

We use a modified version of the reacher domain (Fig. 5) from OpenAI Gym Brockman et al. (2016). It is a two-jointed (2 DOF) robot arm, and the goal is to move the arm's end effector (or '*fingertip*') close to the target location(s). The original environment spawn a target at a random position. For our purposes, we use a modified version with four additional target positions denoted by colored balls to induce subgoals for complex trajectories.

**Tasks** Four tasks were designed to satisfy four logical conditions of increasing difficulty. **Task OR** requires the agent to touch "*the red or green ball, then the blue ball*". **Task IF** stipulates to touch "*the blue ball, unless cancel; then go to the red ball*". The modified environment generates a random 'cancel' event to facilitate this task. **Task Sequential** is equivalent to a patrol task which requires the agent to touch "*the red, green, blue and yellow ball in order*". Lastly, **Task Composite** is a hybrid task that composes all the preceding logics by requiring the agent to "*go to red or green, then blue, unless cancel, then go to yellow*". Figure 6 shows the (perfect) reward structure for the four tasks along with the task name and the natural language description of the task. The tasks are presented in order of increasing complexity.

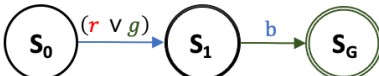

(a) **OR**: Touch the red or green ball, then the blue ball

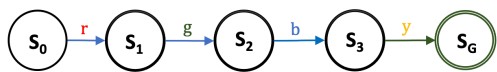

(b) **Sequential**: Touch the red, green, blue and yellow ball in order

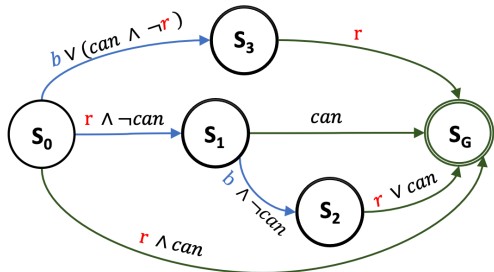

(c) **IF**: Touch blue, unless *cancel*; then touch the red ball

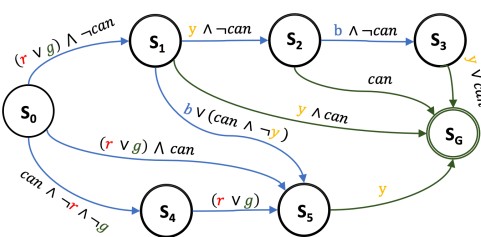

(d) **Composite**: Touch red or green, then blue, unless *cancel*, then go to yellow

Figure 6: The four increasingly difficult tasks (`OR, Sequential, IF, Composite`) with the corresponding task descriptions and FSA states

**Generating Expert Trajectories** Expert trajectories were generated by training a **f-irl** Ni et al. (2020) model with forward KL divergence (fkl) with perfect reward structure using SAC Haarnoja et al. (2018). Implementation details of the expert training and trajectories are detailed in Appendix E.3. 16 expert trajectories generated by the fully trained expert were used for carrying out experiments in this section.

**Results** Figure 7 shows the training curves of our system and baselines. The baselines saturate at a partial reward level, given for reaching the end location; however, learning the optimal reward requires learning to satisfy prior conditions (specific order as stipulated by the task's logical constraints), which none of the baselines are able to recover. With our approach, the agent successfully learns the optimal reward function in all 3 tasks barring the last and

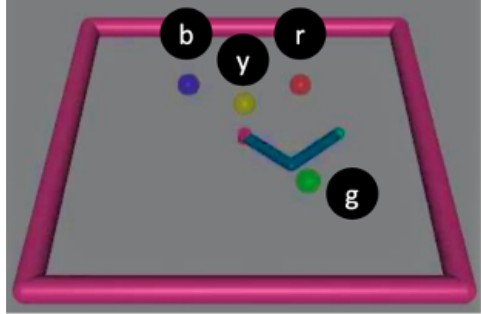

Figure 5: ReacherDelivery-v0: the colored balls specify the various target locations, with the black circles depicting corresponding labels. The arm's *fingertip* represents the agent's location.

most difficult composite task. The perfect FSA structure for Task 4 is 7 states with complex logical constraints among them. We suspect that our model converges on an erroneous sub-optimal reward structure while learning a RM with large state space. Thus, it underperforms against sub-optimal baseline agents in the composite task.

## 5 LIMITATIONS & FUTURE WORK

Comprehensive exploration of IRL in POMDPs is beyond the scope of our work. In general, it is intrinsically an ill-posed problem (due to the nature of IRL) and also can be computationally intractable due to the difficulty in solving POMDPs. Choi & Kim Choi et al. included a detailed exploration in this setting, and future work might compare our model's performance against the three approaches that use sampled trajectories (Max-margin between values (MMV), Max-Margin between feature expectations (MMFE), and the projection (PRJ) methods) and corresponding domain experiments. However, we notice that these methods are derivatives or extensions of algorithms outlined in Abbeel & Ng (2004); Ng et al. (2000), therefore they may have been subsumed by our baselines used, making a proxy comparison with these approaches.

A limitation of our model is the intrinsic limitation or weakness of using RM (or FSA-based abstractions) as structural motifs. While we don't handcraft domain specific structures (they are learnt), but

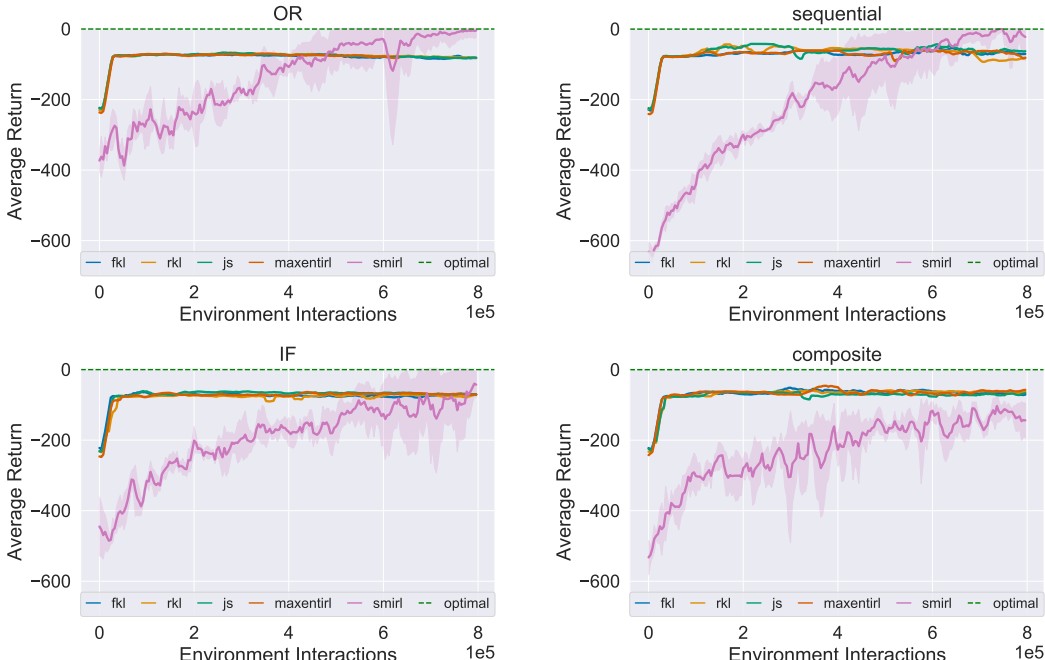

Figure 7: Training curves for our system (SMIRL) and baselines on the **ReacherDelivery** environment. Solid curves and shaded areas depict the mean and standard deviation of three trials with different random seeds respectively for each of the four tasks.

the learning process itself is dependent on a domain-specific labelling function $\mathcal{L}$ assumption. Such perfect domain sensing may not be generalize to all environments.

# 6 BROADER IMPACT AND CONCLUSION

In this work, we proposed a novel approach to learn an agent's reward function from expert observations (i.e., the IRL problem) by inducing structural motifs (in the form of learned reward machines). We empirically show that our approach (SMIRL) is able to successfully learn complex (non-Markovian) sequential tasks in both discrete grid world and high-dimensional continuous domain environments, where foundational and SOTA IRL methods fail. These results are highly motivating for further applications in continuous domain like robotics. Especially, our approach is highly promising for the RoboNLP domain, with the vast swath of rich literature on parsing logical structures from natural language, thus enabling the inverse mapping of agent (robot) behavior to natural language descriptions.

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

# Appendices

## A    DEFINITIONS

> **Definition 1.1: Domain vocabulary & labeling function**
>
> A *vocabulary* is a set of environment or domain specific (boolean) propositional symbols, $\mathcal{P}$, with $\mathbb{R} \in [0, 1]$ for each state $s \in \mathcal{S}$. A *labeling function* is a function $\mathcal{F}_L(t) : \mathcal{S} \times \mathcal{A} \times \mathcal{S}' \to 2^{\mathcal{P}}$, that maps state observations to truth assignments over the vocabulary $\mathcal{P}$ for the time-step $t$.

> **Definition 1.2: MDP with a reward machine (MDP-RM)**
>
> A Markov decision process with a reward machine is a tuple $\Psi = \langle S, A, p, \gamma, \mathcal{P}, \mathcal{F}_L, U, u_0, G, \delta_u, \delta_r \rangle$ where $S, A, p$, and $\gamma$ are defined as in an MDP, $\mathcal{P}$ is a set of domain specific propositional symbols, $\mathcal{F}_L$ is a labelling function (Def. 1.1), and $U, u_0, G, \delta_u$ and $\delta_r$ are defined as in a reward machine, where U is a set of FSA states, $u_0$ is the initial state, $G$ is a set of terminal or goal states, and $\delta_u, \delta_r$ are state and reward transition functions.

## B    REWARD MACHINE: EXPANDED BACKGROUND AND MOTIVATION

**Reward machines** were introduced by Icarte et al. Icarte et al. (2018) as a type of finite state machine (FSM) that supports the specification of reward functions while exposing reward function structure. As a form of FSM, reward machines have the expressive power of a regular language and as such, support loops, sequences and conditionals. Additionally, it supports expression of temporally extended linear-temporal-logic (LTL) and non-Markovian reward specification, where the underlying reward received by an agent from the environment is not Markovian with respect to the state.

When an environment dynamics is specified using a reward machine (RM), as an agent acts in the environment, moving from state to state, it also moves from state to state within a reward machine (as determined by high-level events detected within the environment).

After every transition, the reward machine outputs the reward function the agent should use at that time. For example, we might construct a reward machine for '*delivering mail to an office*' using two states. In the first state, the agent does not receive any rewards, but it moves to the second state whenever it gets the mail. In the second state, the agent gets rewards after delivering the mail. Intuitively, defining rewards this way improves *scale efficiency* as the agent knows that the problem consists of two stages and might use this information to speed up learning.

Using the above discussion, we can define a standard RM using mathematical formalism as the following definition 2.1.

> **Definition 2.1: A standard Reward Machine (RM)**
>
> Given a set of propositional symbols $\mathcal{P}$, a set of (environment) states $S$, and a set of actions $A$, a reward machine is a tuple: $\mathcal{R}_{\mathcal{PSA}} = \langle U, u_0, F, \delta_u, \delta_r \rangle$ where $U$ is a finite set of states $u_0 \in U$ is an initial state, $F$ is a finite set of terminal states (where $U \cap F = \emptyset$), $\delta_u$ is the state-transition function s.t. $\delta_u : U \times 2^{\mathcal{P}} \to U \cup F$, and $\delta_r$ is the state-reward function, s.t. $\delta_r : U \to [S \times A \times S \to \mathbb{R}]$.

### B.1    MOTIVATION FOR USING RM IN IRL+POMDP SETTING

Agents in modern, real-world RL datasets (e.g. robotics, embodied-ai Shridhar et al. (2020)) often, if not always, are required to perform tasks that are long-horizon with compositional and/or logical underlying reward structure. The main motivation of SMIRL is to show that imbuing the agent with *apriori* (approximate) structure of the latent reward associated with a task allows solving complex tasks that are hard to learn from only demonstrative expert trajectory data. This motivation makes the IRL+POMDP problem setting ideal for purposes in this paper.

The effectiveness of automata-based memory has long been recognized in the POMDP literature Cassandra et al. (1994), where the objective is to find policies given a complete specification of the environment. The overarching idea in approaches under this umbrella is to encode policies using Finite State Controllers (FSCs), which are FSMs with states associated with one primitive action, and the transitions are defined in terms of low-level observations from the environment. During environment interaction, the agent always selects the action associated with the current state in the FSC. Using such automata-based memory was leveraged to work in the RL setting in work by Meuleau et al. Meuleau et al. (2013b) by exploiting policy gradient to learn policies encoded as FSCs.

RMs can be considered as a generalization of FSC as they allow for transitions using conditions over high-level events and associate complete policies (instead of just one primitive action) to each state. This allows RMs to leverage existing deep RL methods to learn policies from low-level inputs, such as images, which is not achievable by other automata-based approaches like Meuleau et al. (2013b). That said, learning FSMs using other ontologies (e.g. Xu et al. (2020); Zhang et al. (2019)) do exist in concurrent literature. Discussion and delineation with such works are further discussed in the 'Related Work' section (S. C).

## C  RELATED WORK

Augmenting memory using of Recurrent Neural Networks (RNNs) in combination with policy gradient Jaderberg et al. (2016); Mnih et al. (2016); Schulman et al. (2017) is a common approach in state-of-the-art (SOTA) approaches in the RL+POMDP domain. Other approaches use external neural-based memories Oh et al. (2016); Khan et al. (2017); Hung et al. (2019). Model-Based Bayesian RL and extension approaches Doshi-Velez et al. (2013); Ghavamzadeh et al. (2015); Poupart & Vlassis (2008) under partial observability provide a small binary memory to the agent and a special set of actions to modify it. The motivation and idea behind our work here are largely orthogonal to these aforementioned approaches.

The work that is closest to ours is by Icarte et al. Toro Icarte et al. (2019) – where the authors learn RMs for partially observable RL tasks from trajectories. However, efficacy of the approach is shown in 2D discrete domains, with the authors noting the challenge of showcasing them in 3D continuous domain due to the intractability of state space explosion. While Toro Icarte et al. (2019) motivates our choice to use RM as the chosen structural motif architecture (as opposed to other available FSA ontologies), but to the best our knowledge, learning the motif on continuous 3D domains with complex logic (see SMIRL Algorithm) is not undertaken in prior works. While this difference can be argued as meagre for 2D toy like domains, but is significant for continuous domains, because 'Tabu search' for state space is computationally infeasible in such complex domains. Thus, this insight is more applicable in solving IRL in real-world robotic or embodied-ai domains commensurate with the motivation of our work.

Another related sub-domain of works include literature on *learning logic and automata from demonstrations*. These works by problem definition is slightly different to the IRL problem domain we tackle here. The works in this area (e.g. by Vazquez et al. Vazquez-Chanlatte et al. (2018; 2021)) infers Boolean non-Markovian rewards, or logical properties of available traces (aka. demonstrations). This is achieved by learning probabilistic densities of demonstrations over an **existing, apriori** knowledge pool of candidate specifications. In essence, it is a specifications matching problem, or searching for the most probable specification in a pool of candidate specifications. Our work is **orthogonal to these** in the aspect that we do not have or define the task labels or any apriori structure of the specification.

## D  OFFICE GRIDWORLD DOMAIN

### D.1  DETAILED ILLUSTRATION OF STRUCTURAL MOTIF LEARNING (SMIRL)

Here we examine and illustrate the SMIRL learning algorithm 1 using *Task 3: 'Fetch and deliver coffee and mail'*. The Fig. 8 juxtaposes the perfect reward structure with a FSA structural motif learned using the SMIRL 1 algorithm. Although the learned structure is not optimal (with 6 FSA states as opposed to optimal 4), but if converges to the desired target state.

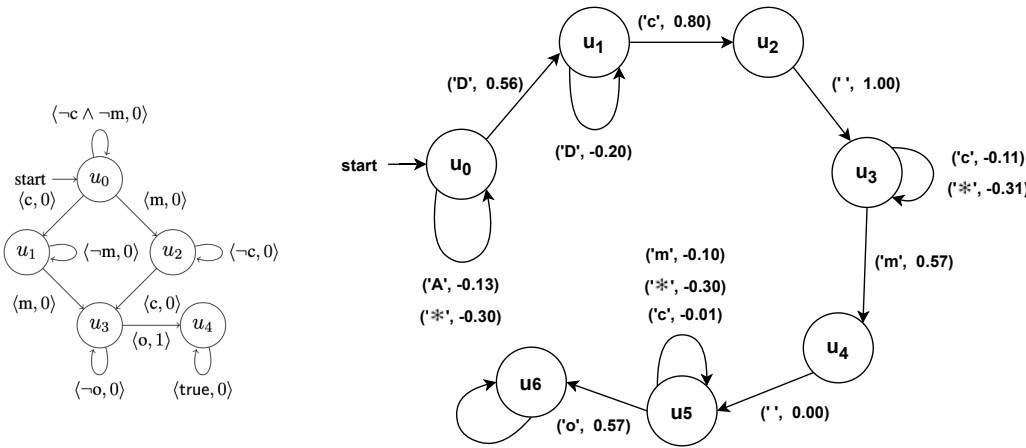

(a) State diagram of a **perfect** FSA

(b) State diagram of **learned** FSA

Figure 8: Qualitative Evaluation in the *Office GridWorld*. **Left**: Fig. 8a shows the **perfect** reward (FSA) structure for *Task 3: Fetch & Deliver 'Coffee' and 'Mail'* requiring the delivery of both coffee ('c') and mail ('m') to the office ('o') starting with position 'A' on the map and initial FSA state $u_0$. **Right**: Fig. 8b shows the **learned** FSA structural motif and weights. The state transition arrow labels (e.g. $u_1 \rightarrow u_2$: ('c', 0.80) )indicate the true propositional symbols [Def. 1.1] – from an agent action $a$ resulting in state transition from $s$ to $s'$, and the underlying FSA to state $u' = \delta_u(u, \mathcal{F}_L(s, a, s'))$ – and the correponding reward $r(s, a, s')$, where $r = \delta_r(u)$. We can see some artifacts of the SMIRL algorithm 1 here with nodes $u_2$ and $u_4$. The first transition from $u_0$ to $u_1$ comes from the fact that the expert trajectories go through D while fetching coffee c.

---

**Lemma 1: MDP, MDP-RM expected reward equivalency**

Given an MDP-RM $\Psi = \langle S, A, p, \gamma, \mathcal{P}, \mathcal{F}_L, U, u_0, G, \delta_u, \delta_r \rangle$ let $\mathcal{M}_\Psi = \langle S', A', r', p', \gamma' \rangle$ be the MDP defined such that $S' = S \times U, A' = A, \gamma' = \gamma,\ p'(\langle s', u' \rangle \mid \langle s, u \rangle, a) = \begin{cases} p(s' \mid s, a) & \text{if } u' = \delta_u(u, \mathcal{F}_L(s')) \\ 0 & \text{otherwise} \end{cases}$, and $r'(\langle s, u \rangle, a, \langle s', u' \rangle) = \delta_r(u, u')(s, a, s')$.

Then any policy for $\mathcal{M}_\Psi$ achieves the same expected reward in $\Psi$, and vice versa.

---

### D.2 IMPLEMENTATION DETAILS

**Generating expert trajectories** The process flow for generating expert trajectories in the gridworld domain entails first to train an expert policy, $\pi_e$ using a perfect FSA reward structure (Fig. 8a), then using the expert policy to generate $\mathcal{D}_e$

The final form of the expert demonstrations is represented by series of state-action transitions, i.e., $(s_1, a_1, s_2, a_2, ..., s_T)$. The actions are represented by single integers from 0 to 3. For the low-level state representations, we include both high-level features of the environment (one-hot encoding of one of the high-level positions, e.g., {A, B, ..., D, c, m, ..} etc.) and the low-level positions (one-hot encoding of the 108 grids of the *Office GridWorld* environment).

The following points detail various conditions adhered to while $\mathcal{D}_e$ generation:

1. There are K max steps (episode horizon) possible in an episode.

2. An episode can end in (t « K) steps if a 'done' or 'game over' condition is hit (like stepping on obstacles).

3. Once an optimal trajectory is traversed for 1 round trip (e.g. *Task 4 – 'Patrol ABCD'*: $u_0 \rightarrow u_1 \rightarrow u_2 \rightarrow u_3$), the agent receives a reward of 1.

4. After that, the agent takes k random steps.

5. If not terminated in k steps, `get_optimal_action()` is invoked (from whichever random position the agent is in, creating another successful reward trip completion.

6. The above step is repeated until max K steps are reached

## E  REACHER DOMAIN DETAILS

In this section we present pertinent details about the experiments conducted on the continuous MuJoCo Reacher domain (Sec. 4.3).

### E.1  REACHERDELIVERY-V0: MODIFIED REACHER DOMAIN

We modify the classic OpenAI Gym's Brockman et al. (2016) MuJoCo Todorov et al. (2012) Reacher environment to our purposes here. The original Reacher environment is a two-jointed robot arm, and the goal is to move the robot's end effector (called *fingertip*) close to a target that is spawned at a random position. The *action space* consists of an *action* $(a, b)$ that represents the torques applied at the hinge joints. The *observation space* consists of the sine, cosine angles of the two arms, coordinates of the target, angular velocities of the arms and a 3D distance vector between the target and the reacher's fingertip. The *reward* consists of two parts: i. *reward_distance* ($R_d$): a measure of how far the fingertip of the reacher (the unattached end) is from the target, with a more negative value assigned with increasing distance; ii. *reward_control* ($R_c$): a negative reward for penalising actions that are too large. It is measured as the negative squared Euclidean norm of the action, i.e. as $-\sum action^2$. The total reward returned is: $reward = reward\_distance + reward\_control$ ($R = R_d + R_c$).

**ReacherDelivery**   The following code snippet shows how the MuJoCo asset file was modified to add four target locations as colored balls.

Excerpt from `reacher_delivery.xml` file showing added red ball as a target location

```
<!-- RED -->
<body name="red" pos="0 0 0.01">
      <joint armature="0" axis="1 0 0" damping="0" limited="true" name="
          red_x" pos="0 0 0" range="-.27 .27" ref="0" stiffness="0" type
          ="slide"/>
      <joint armature="0" axis="0 1 0" damping="0" limited="true" name="
          red_y" pos="0 0 0" range="-.27 .27" ref="0" stiffness="0" type
          ="slide"/>
      <geom conaffinity="0" contype="0" name="red" pos="0 0 0" rgba="0.9
          0. 0. 0.3" size=".02" type="sphere"/>
</body>
```

We extended the MuJoCo Reacher class with an environment MDP wrapper and added target goals. Following an agent step in the environment, the state proposition labels are updated using the labelling function. The following code snippet exemplifies this, where a target color location proposition is labeled 'true' if the fingertip is within a threshold distance from it.

The labelling function $\mathcal{F}_L$ for `ReacherDelivery` domain

```
true_props = []
if dist_red < 0.02:
    true_props.append('r')
if dist_green < 0.02:
    true_props.append('g')
if dist_blue < 0.02:
    true_props.append('b')
if dist_yellow < 0.02:
    true_props.append('y')
if cancel:
    true_props.append('c')
```

### E.2  TASKS REWARD FSA STRUCTURES

Figure 9 shows the (perfect) reward structure for the four tasks along with the task name and the natural language description of the task. The tasks are presented in order of increasing complexity.

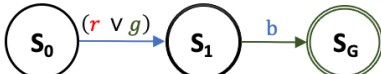

(a) **OR**: Touch the red or green ball, then the blue ball

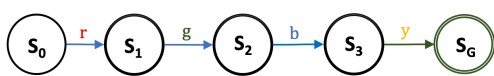

(b) **Sequential**: Touch the red, green, blue and yellow ball in order

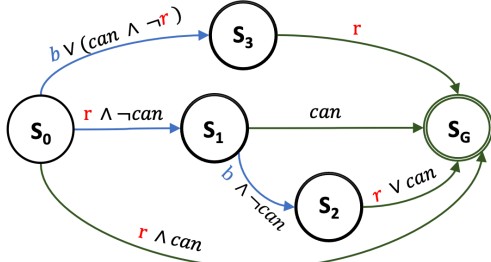

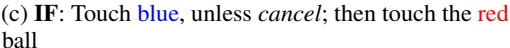

(c) **IF**: Touch blue, unless *cancel*; then touch the red ball

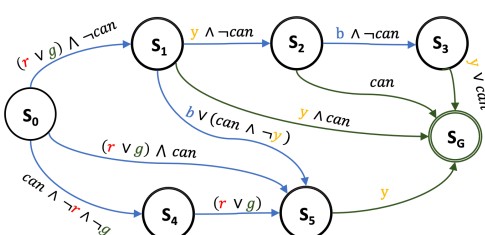

(d) **Composite**: Touch red or green, then blue, unless *cancel*, then go to yellow

Figure 9: The four increasingly difficult tasks (OR, Sequential, IF, Composite) with the corresponding task descriptions and FSA states

### E.3 IMPLEMENTATION DETAILS

This section outlines the various implementation details for the *ReacherDelivery* domain experiments (Sec. 4.3).

**Training Details:** We use SAC as the underlying RL algorithm throughout. The policy network is a tanh squashed Gaussian with mean and standard deviation parameterized by a (64, 64) ReLU MLP with two output heads. The Q-network is a (64,64) ReLU MLP. For optimization we use Adam with learning rate of 0.003 for both the Q-network and policy network. The replay buffer size was 10000 and we used batch size of 256.

For the baselines f-IRL and MaxEntIRL, we used the f-IRL Ni et al. (2020) authors' official implementation[2]. f-IRL and MaxEntIRL require an estimation of the agent state density. We use kernel density estimation to fit the agent's density, using Epanechnikov kernel with a bandwidth of 0.2 for pointmass, and a bandwidth of 0.02 for Reacher. At each epoch, we sample 1000 trajectories (30000 states) from the trained SAC to fit the kernel density model.

**Generating Expert Trajectories:** For expert trajectories generation, we first train expert policies imbued with perfect reward structure using SAC for each of the tasks. Fig. 10 shows the training curve and violin plot expert return density curves of training.

SAC uses the same policy and critic networks with the learning rate set to 0.003. We train using a batch size of 100, a replay buffer of size 1 million, and set the temperature parameter $\alpha$ to be 0.2. The policy is trained for 1 million timesteps on ReacherDelivery. All algorithms are tested on 16 trajectories collected from the expert stochastic policy.

**Evaluation** We compare the trained policies by the baselines (f-IRL< MaxEntIRL) and SMIRL by computing their returns according to the ground truth return on the ReacherDelivery environment.

**Computational Complexity** In general, *ceteris paribus*, SMIRL is more sample efficient (i.e. converges to a solution faster) than baseline IRL approaches. Complexity can arise from two factors: i. the domain complexity, say 2D discrete vs. continuous, and ii. The size and complexity of the reward machine (or, FSA) structural motif. We see implications of both in the paper. First, the sample complexity increases with increasing domain complexity (office gridworld vs. Reacher) and this is intuitive. From our experiments, for the second kind of complexity, i.e. with more intricate underlying RM structure, the bottle-neck seems to emanate from the ability to learn the structure. We see that for the hardest (composite) task, the structure was not learned, and increasing sample complexity wouldn't have helped (it would saturate to a suboptimal level). If the structure is learnable, then

---

[2]https://github.com/twni2016/f-IRL

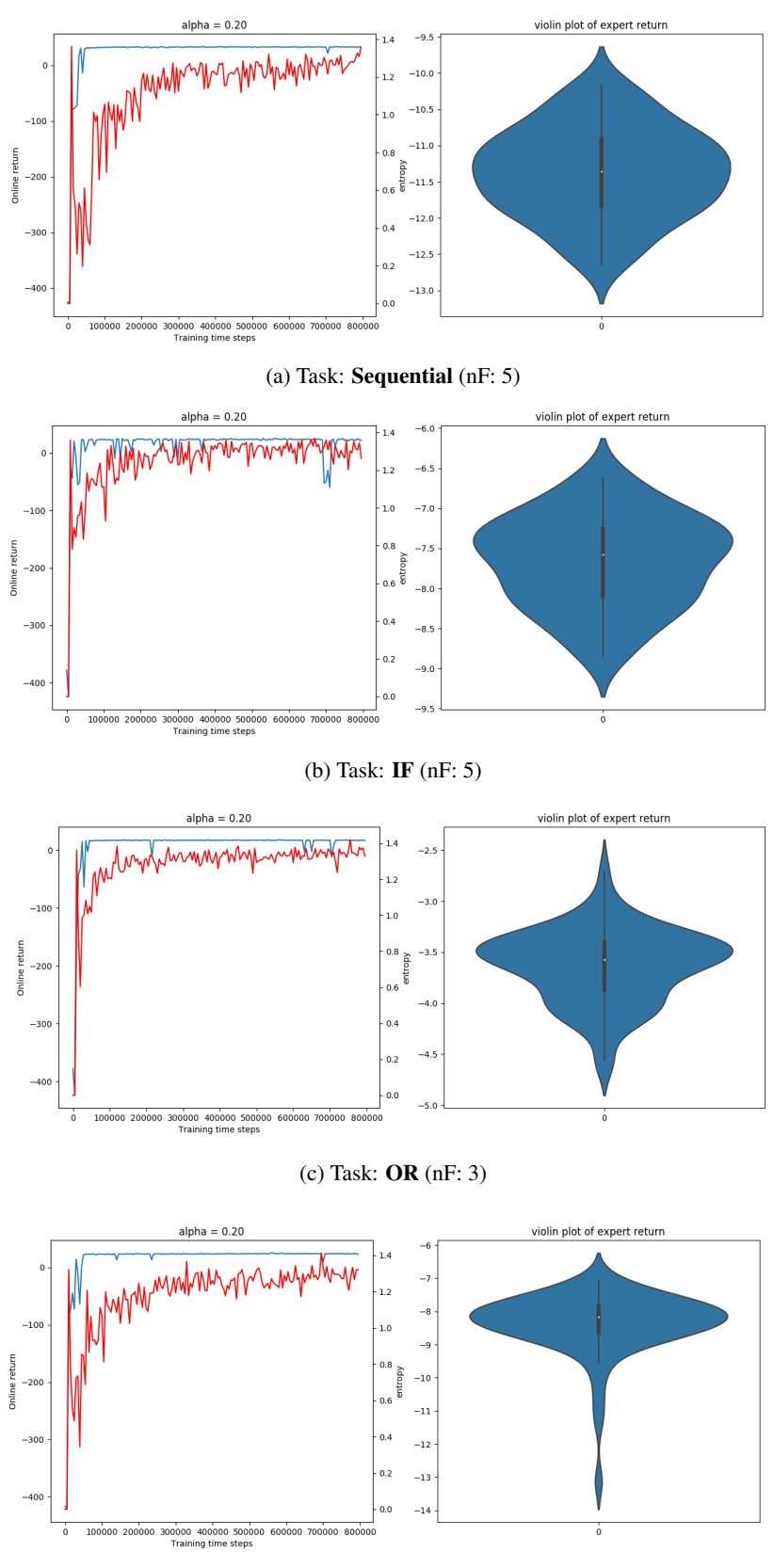

(a) Task: **Sequential** (nF: 5)

(b) Task: **IF** (nF: 5)

(c) Task: **OR** (nF: 3)

(d) Task: **Composite** (nF: 7)

Figure 10: The training curves for expert policy training. The experts were imbued with perfect state based reward structure, where the number of states are given within brackets as 'nF' value. **Left**: The blue curve shows reward and the red curve shows average entropy. **Right**: The violin plot of expert return density.

sample complexity scales with the complexity of the task structure (e.g. 'sequential' vs. 'OR' in Fig. 6).

## F    BASELINE OBJECTIVE FUNCTIONS

**f-IRL**    We train the three variants of f-IRL: forward KL (fkl), reverse KL (rkl) and Jansen-Shannon (js) that represents the f-divergence metric used by the f-IRL algorithm.

f-divergence Ali & Silvey (1966) is a family of distribution divergence metric, which generalizes forward/reverse KL divergence. Formally, let P and Q be two probability distributions over a space $\Omega$, then for a convex and Lipschitz continuous function $f$ such that $f(1) = 0$, the f-divergence of $P$ from $Q$ is defined as:

$$D_f(P\|Q) := \int_\Omega f\left(\frac{dP}{dQ}\right) dQ \qquad (10)$$

f-IRL uses state marginal matching by minimizing the f-divergence objective:

$$L_f(\theta) = D_f\left(\rho_E(s)\|\rho_\theta(s)\right) \qquad (11)$$

This objective is realized by computing the exact (analytical) gradient of the preceding f-divergence objective w.r.t. the reward parameters $\theta$.

---

**Theorem 6.1: f-divergence analytic gradient (from Ni et al. (2020))**

The analytic gradient of the $f$-divergence $L_f(\theta)$ between state marginals of the expert ($\rho_E$) and the soft-optimal agent w.r.t. the reward parameters $\theta$ is given by:

$$\nabla_\theta L_f(\theta) = \frac{1}{\alpha T} \operatorname{cov}_{\tau \sim \rho_\theta(\tau)} \left(\sum_{t=1}^{T} h_f\left(\frac{\rho_E(s_t)}{\rho_\theta(s_t)}\right), \sum_{t=1}^{T} \nabla_\theta r_\theta(s_t)\right)$$

where $h_f(u) \triangleq f(u) - f'(u)u$, $\rho_E(s)$ is the expert state marginal and $\rho_\theta(s)$ is the state marginal of the soft-optimal agent under the reward function $r_\theta$, and the covariance is taken under the agent's trajectory distribution $\rho_\theta(\tau)$.[2]

---

