# OpenReview forum: "Learning then Leveraging Structures Help with Complex, Compositional, Causal Sequential Tasks in Inverse Reinforcement Learning"
_ICLR.cc/2024/Conference — ICLR 2024 Conference Withdrawn Submission_

### Official Review · Reviewer_aQAV · 2023-10-28

**Soundness:** 2 fair
**Presentation:** 2 fair
**Contribution:** 2 fair
**Rating:** 3
**Confidence:** 3

**Summary:**

The paper addresses inverse RL tasks that contain a behavioral strategy that is partially observable in that sense that it is sequential in the goals that are to be achieved. The proposed approach first learns a task structure in the form of a finite state automaton and then attempts to solve the IRL problem within the learned task structure. It is clearly shown that with a suitable task structure partially observable policies can be inferred – which is, however, not overly surprising.

**Strengths:**

An intricate combination of two learning challenges – task structure and reward function learning. I was not overly familiar with the challenge that lies in such a combination. The challenge is definitely interesting and seems unsolved at the moment.

Performance in the proposed somewhat toy problems is clearly better than approaches that cannot learn complex partially observable policies.

The formal background is appealing.

**Weaknesses:**

From a maybe slightly simplistic view, the paper essentially states that if you do not enable an algorithm to represent more complex sequential behavioral policies, then you will not learn about them. To the best of my understanding this is the story of Figure 4… as a result, I am not overly convinced of the strength of this work.

The learning of the FSA structure (or of an approximation of such a structure by any other means) seems not to be really addressed in any detail. This is highly confusing for me because in my opinion this is the true challenge. Once you have it and test on a problem that requires it, then clearly it will work better.

Generally, the paper introduces POMDP and other formalisms in solid detail. I partially can imagine moving some of these parts to the appendix. They do not really reveal what the authors have actually done in the end.

I went astray on how the FSA model is learned. What is the actual input given to the learner to learn the FSA model, and how is it then actually used to infer the reward function. Given a model such as the ones depicted in Figure 2b and c, then I can well imagine how this is done. The model learning is hardest. The authors here write about “Hand-crafted” features in the reward function learning subsection. The RM structure learning is referred to other work without providing any intuition.

I am also fully left astray what the actual observations / state representations are that you feed into your model. It, for example, (as you must well-know) makes a huge difference if you have the possible state space S given a priori or if you really only have access to observations O without any prior knowledge of the potential structure of S.

Seeing these weaknesses, I currently cannot recommend publication. But, to be honest, I am at this point rather uncertain about the details and their potential merit.

**Questions:**

A few more concrete questions (but see my main concerns under weaknesses above for the main issues):

I am not quite familiar with the concept of “propositions” and a corresponding “labelling function”. I think a brief running example would help the reader a lot throughout the paper. This could help foster intuition about propositions and other formal concepts denote.

Algorithm 1 starts with “Learn a FSA structure” … isn’t this the main challenge – how come there are not any more details on this?

Where does learning the RM – and particularly also Eq. 3 – become relevant in Algorithm 1?

How do the actual FMs look like that were learned? Do they 1:1 fit with the once underlying the behaviors of the expert trajectories? What happens when they do not fit 1:1?

What are the actual state spaces and observations in the respective experiments?

---

### Official Review · Reviewer_eYKt · 2023-10-31

**Soundness:** 2 fair
**Presentation:** 3 good
**Contribution:** 2 fair
**Rating:** 5
**Confidence:** 4

**Summary:**

This paper proposes to first construct a reward machine (RM) from IRL datasets and then learn reward functions from the constructed reward machine (RM). The authors suppose that grounding IRL methods with reward machines, which encode the underlying relationships between actions and outcomes, could enhance the performance.

**Strengths:**

(1) The paper noticed the potential rich structure of the reward functions and make use of that in solving IRL problems.
(2) The proposed method is logically clear and straightforward.

**Weaknesses:**

(1) The proposed method is a combination of RM learning algorithms from established work and the well-known IRL algorithm MaxEnt IRL. I would expect more efficient learning algorithms once you have access to the RM instead of simply augmenting the trajectory with RM states and apply MaxEntRL again.
(2) There is already a rich line of IRL research that doesn’t assume access to the true underlying dynamics, e.g.,
Guided Cost Learning: Deep Inverse Optimal Control via Policy Optimization from ICML 16'. I am afraid for the problem setting considered in this work it is hardly beneficial enough to introduce the extra complexity of using RMs. If the author could enhance the proposed method to allow learning in high-dimensional, continuous domains, that will make it a stronger submission.
(3) When you claim you learn the “causal structure” of the problem, I am expecting to see some discussion using some causal formalisms like SCMs or potential outcomes. In this paper, the theory does not contain any causal analysis so I would suggest the author not claim “causal” in the main text.
(4) From the experiments, I cannot see clearly the advantage of having RM in IRL. The only comparison that SMIRL outperforms the baselines by a big margin is a POMDP environment while those baselines are not designed to work in POMDP at all. When in MDP environments, the proposed method doesn’t significantly outperform the baselines which damps the effectiveness of the proposed method.

**Questions:**

(1) Could you elaborate more on the accuracy of the learned RM compared against the ground truth RM? I see in the supp material that the learned RM is far away from the ground truth. What are the true causes would you think that such a seemingly inaccurate learned RM is still useful to the learning?
(2) In preliminaries, section "IRL with Partial observability", the author mentioned that "unlike our algorithm, ..., (another work) assumed full access to the environment transition probabilities..." claiming that this work is not assuming known dynamics. However, at the beginning of section 3, the author stated that "we assume that we have access to $\mathcal{M}\setminus \{R\}$ (the transition dynamics)...". Is this a contradiction? And as far as I know, MaxEnt IRL needs known dynamics to work.

---

### Official Review · Reviewer_XJYd · 2023-11-01

**Soundness:** 2 fair
**Presentation:** 2 fair
**Contribution:** 2 fair
**Rating:** 3
**Confidence:** 4

**Summary:**

This paper presents a method, SMIRL (Structural Motif-Conditioned Inverse Reinforcement Learning), for identifying causal, logical reward functions represented as Reward Machines from demonstration data, and then using Maximum entropy inverse reinforcement learning to identify a specific reward function given this reward machine representation. The method assumes access to logical propositions that accurately describe each state within the environment. SMIRL is demonstrated on several tasks including a 2D discrete grid world environment requiring navigation, and a 3D continuous manipulation task requiring reaching different points in 3D space.

**Strengths:**

This paper tackles a common problem in inverse reinforcement learning: how to deal with partially observable settings. This topic is crucial for real-world setups where robots are very unlikely to have full information about the environment when trying to perform IRL. As a result, I expect many people at ICLR to be interested in this topic.

The method is presented on multiple environments, both simpler 2D discrete settings, and more complex, continuous 3D settings. SMIRL’s strengths are clearest in the 2D discrete case, but SMIRL also outperforms baselines on a subset of the 3D continuous tasks as well. It is helpful to see comparisons to the foundational methods of apprenticeship learning and maximum entropy IRL, which perform poorly on the 2D setup. It is similarly helpful to see a comparison to f-IRL for the continuous domain (js and rkl are also compared, but not defined in the main text. In the Appendix it is mentioned that they are variants of f-IRL). The appendix is substantial and includes a lot of extra work and information.

**Weaknesses:**

Fundamentally, while interesting, I do not believe that ICLR is the right venue for SMIRL. ICLR is a conference where “learning representations” is part of the name, and SMIRL fundamentally relies on fixed representations with propositional logic. The foundations for SMIRL, which is the reward machine approach, are consistently published in AI venues such as JAIR, IJCAI or AAAI because the audiences there are more appreciative of these approaches, and have the background to better appreciate this kind of work. My comments below are therefore intended to help improve the clarity of the paper, but I believe it would be better suited to a different venue.

### Unclear methodological contribution, restructuring of related work / background needed ###
The paper has an unusual structure for a machine learning paper, where the related work and background section are not separated. They are instead joined under “Preliminaries and background”. This section is not well structured, and it is therefore very difficult to determine what is novel about the presented method.

For example, the first three subsections of “Preliminaries and background” are obviously applicable to all IRL models, and so that seems appropriate. However, diving into MaxEnt IRL, there is already some confusion, as SMIRL relies on MaxEnt IRL as a sub-routine, but also distinguishes itself from MaxEnt IRL by using a Reward Machine (from my understanding based on reading the rest of the paper). Similarly, the next section on “IRL with Partial Observability” mentions a separate approach from Choi and Kim, but which has no connection to the SMIRL method (other than being a piece of related work, but it is not a preliminary). The final paragraph returns to Reward Machine, which SMIRL is based on, but slightly different from, in ways that are not clear from reading these sections.

To help a reader understand what the new methodological contribution is, there should be separate “background” and “related work” sections, that clearly distinguish SMIRL from alternative approaches. Otherwise it is very difficult to determine how it is different from, e.g. Icarte et al 2018, or MaxEnt IRL.

To further demonstrate this point, in the Appendix the paper states: ““... to the best our knowledge, learning the motif on continuous 3D domains with complex logic (see SMIRL Algorithm) is not undertaken in prior works. While this difference can be argued as meagre for 2D toy like domains, but is significant for continuous domains, because ‘Tabu search’ for state space is computationally infeasible in such complex domains. Thus, this insight is more applicable in solving IRL in real-world robotic or embodied-ai domains commensurate with the motivation of our work.”
* It was not at all obvious to me that this was the main point of the paper. Since it is not explained how the SMIRL method differs from the Icarte et al, 2018 work, I could not determine that this was the main methodological novelty. Furthermore, the main method description states that Tabu search is being used (“In order to learn the RM’s structure, we build on the method described in Toro Icarte et al. (2019), where Tabu search Glover & Laguna (1998) is used to compute an RM…”), so I fundamentally misunderstand something about the way the method works. If this is the main methodological difference, the experiments performed should all be on 3D, continuous domains, since that is what differentiates SMIRL from prior work.

### More relevant baselines are needed ###
In the limitations & future work section, the paper notes:
“ Choi & Kim Choi et al. included a detailed exploration in this setting, and future work might compare our model’s performance against the three approaches that use sampled trajectories (Max-margin between values (MMV), Max-Margin between feature expectations (MMFE), and the projection (PRJ) methods) and corresponding domain experiments. However, we notice that these methods are derivatives or extensions of algorithms outlined in Abbeel & Ng (2004); Ng et al. (2000), therefore they may have been subsumed by our baselines used, making a proxy comparison with these approaches.”
* I do not believe this is a good reason for not running relevant baselines. As discussed in the survey paper linked to throughout the paper (Arora & Doshi, 2020), these methods do have important innovations that make them more applicable to POMDP settings. Claiming that these methods are similar enough to a much older approach that was not designed for POMDPs is not reasonable. The paper would be much more informative if it included any comparisons to approaches that were designed for POMDPs, and without this comparison, it is hard to evaluate the importance of SMIRL’s methodological innovation.

### Other smaller comments ###
Almost all citations in the paper are incorrectly formatted in the text, making the paper very difficult to read.

Algorithm 1 is not well formatted. It is not clear from the algorithm box that “learn the reward machine” is actually a comment describing the next few lines of the algorithm, but that is what the text suggests.

The baselines in the continuous control domain are not defined in the main text of the paper.

**Questions:**

* How does SMIRL compare to alternative IRL for POMDP approaches?
* Can the background section be rewritten to better explain SMIRL’s novelty?
* Can a related work section be included to better explain SMIRL’s novelty?
* If the main innovation is to handle 3D continuous scenes, can that be expanded more in the paper?